

# The ratio of serum Angiopoietin-1 to Angiopoietin-2 in patients with cervical cancer is a valuable diagnostic and prognostic biomarker

Ping Yang[1,2], Na Chen[1], Dongyun Yang[1], Janet Crane[3,4], Shouhua Yang[1], Hangyu Wang[5], Ruiqing Dong[6], Xiaoqing Yi[1], Lisha Xie[1], Guo Jing[1], Jing Cai[1] and Zehua Wang[1]

[1] Department of Obstetrics and Gynecology, Union Hospital, Tongji Medical College, Huazhong University of Science and Technology, Wuhan, China
[2] Department of Obstetrics and Gynecology, First Affiliated Hospital, School of Medicine, Shihezi University, Shihezi, China
[3] Department of Orthopaedic Surgery, Johns Hopkins University School of Medicine, Baltimore, MD, United States of America
[4] Department of Pediatrics, Johns Hopkins University, Baltimore, MD, United States of America
[5] School of Pharmacy, Shihezi University, Shihezi, Xinjiang, China
[6] Department of Obstetrics and Gynecology, Tianyou Hospital attended to Wuhan University of Science and Technology, Wuhan, Hubei, China

Corresponding author
Zehua Wang, zehuawang@163.net

## ABSTRACT

**Objectives**. Angiopoietins have been found to play essential roles in tumor angiogenesis. The present study was aimed at investigating the diagnostic and prognostic values of serum angiopoietin 1 and 2 (sAng-1 and sAng-2) in cervical cancer.

**Methods**. The sAng-1 and sAng-2 concentrations were analyzed in 77 women with cervical cancer, 44 women with cervical intraepithelial neoplasia (CIN) and 43 women without cervical lesions by enzyme-linked immunosorbent assay. The diagnostic values of sAng-1, sAng-2 and sAng-1/sAng-2 were evaluated by receiver operating characteristic (ROC) curves. The Ang-1 and Ang-2 expression in cervical cancer tissues as well as microvessel density (MVD), were assessed by immunohistochemistry.

**Results**. The concentration of sAng-2 gradually increased and the sAng-1/Ang-2 ratio was gradually decreased from normal control to CIN, then to squamous cell cancer, and the sAng-1/sAng-2 ratio was also significantly decreased in adenocarcinoma. The area under ROC curves of sAng-2 and sAng-1/sAng-2 ratio for discriminating cervical cancer from normal were 0.744 and 0.705, respectively. Decreased sAng-1/sAng-2 was significantly associated with advanced tumor stage, poor differentiation, lymph-vascular space invasion and high MVD. sAng-2 was positively correlated with the Ang-2 expression in cervix epithelia. A high sAng-1/sAng-2 ratio was associated with a longer progression-free survival and a longer overall survival in cervical cancer patients.

**Conclusions**. These findings suggest that sAng-2 and the sAng-1/sAng-2 ratio may be valuable diagnostic and prognostic biomarkers for cervical cancer.

## INTRODUCTION

Globally, cervical cancer incidence has increased from 378,000 cases per year in 1980 to 528,000 cases per year in 2012, representing a 0.6% annual rate of increase (*Ferlay et al., 2015*). The majority of cervical cancer patients come from economically underdeveloped areas, and they have a higher rate of mortality. Although the cervical screening programs based on the cervical cytology and HPV test can effectively reduce the incidence of cervical cancer, women in underdeveloped areas usually cannot accept standardized screening due to social, religious and psychological factors. Thus, novel circulating biomarkers that allow monitoring of essential molecular events in cervical cancer may improve the detection of lesions that have a high risk of progression in both primary screening and triage settings.

Angiogenesis is essential for tumor progression and has shown great promise as a therapeutic target for treatment of advanced cervical cancer (*Eklund & Saharinen, 2013*; *Kerbel, 2015*). Recently, agents targeting the angiopoietin (Ang)/Tie system have been developed in several clinical trials and achieved encouraging results in anti-angiogenesis therapy (*Gerald et al., 2013*). The human Ang/Tie system is mainly composed of three secreted ligands (Ang-1, Ang-2, and Ang-4) and two tyrosine kinase receptors (Tie-1 and Tie-2). All three human angiopoietins bind directly to Tie-2 and result in the phosphorylation and multimerization of Tie-2, which in turn trigger downstream signaling and regulate vascular remodeling and maturation. Ang-1 functions as a constitutive Tie-2 receptor agonist. The function of Ang-2 is context-dependent; Ang-2 functions as an partial Tie-2 signaling agonist in the presence of Ang-1 or a Tie-2 signaling agonist in the absence of Ang-1 (*Augustin et al., 2009*; *Fagiani & Christofori, 2013*; *Yang et al., 2015*). Ang-1 is widely expressed in periendothelial cells in human quiescent vasculature, where it sustains vessel maturation and stabilization. In contrast, Ang-2 is expressed in endothelial cells during vascular remodeling and angiogenesis, where it mediates vascular destabilization. Thus, the balance of Ang-1 and Ang-2, as well as the cellular context of Tie-2 activation, contribute to the outcome of Tie-2 activity. In addition, angiopoietins have been shown to directly interact with integrins on endothelial, mesenchymal and tumor cells, thereby eliciting Tie-2-independent biological effects (*Augustin et al., 2009*; *Fagiani & Christofori, 2013*; *Shim, Ho & Wong, 2007*). Both Ang-1 and -2 have recently been found to be expressed in tumor cells, and they act cooperatively with VEGF promoting the angiogenesis during tumor progression (*Augustin et al., 2009*; *Ebos & Kerbel, 2011*; *Gerald et al., 2013*; *Shim, Ho & Wong, 2007*; *Yang et al., 2015*).

Increased circulating serum Ang-2 (sAng-2) concentrations or a shift in the sAng-1/sAng-2 ratio in favor of sAng-2 in tumor patients has been shown to correlate with advanced tumor progression and poor survival in epithelial ovarian cancer (*Sallinen et al., 2014*; *Sallinen et al., 2010*), melanoma (*Helfrich et al., 2009*), metastatic colorectal cancer (*Goede et al., 2010*), neuroendocrine tumors (*Detjen et al., 2010*), hepatocellular carcinoma (*Bouattour, Payance & Wassermann, 2015*) and pancreatic cancer (*Schulz et al., 2011*). Furthermore, a soluble Tie-2 receptor can sequester angiopoietins (*Augustin et al., 2009*; *Schulz et al., 2011*). In cervical cancer, based on a preliminary study on plasma concentrations of Ang-1and Ang-2 in 34 patients with cervical cancer, circulating Ang-1,

Ang-2, and Tie-2 and the Ang-1/Ang-2 ratio were significantly increased compared with healthy women (*Kopczynska et al., 2009*). Moreover, decreased Ang-1/Ang-2 ratio was associated with advanced Federation of Gynecology and Obstetrics (FIGO) stage (*Kopczynska et al., 2009*). However, the diagnostic and predictive value of circulating Ang-1 and Ang-2 in cervical cancer remains largely unclear.

In the present study, the diagnosis and prognostic value of sAng-1, sAng-2 and sAng-1/sAng-2 in patients with cervical cancer was evaluated. In addition, the correlations between the circulating angiopoietin concentrations and the angiopoietin expression levels and microvessel density (MVD) in tumor were investigated.

## MATERIALS AND METHODS

### Study groups and inclusion criteria

We conducted a non-matched case-control study using serum samples obtained from patients who received treatment at Wuhan Union Hospital (Wuhan, Hubei, China) between February 2012 to March 2014, including 44 patients with CIN, 77 patients with cervical cancer (61 squamous cell carcinomas, CSCC, and 16 adenocarcinoma, CADC) and 43 patients with benign disorders of uterus such as myoma or adenomyosis as controls. The diagnoses were pathologically verified and patients with medical diseases such as other type of cancer or inflammatory, atherosclerotic and connective tissue disease were excluded. The age range of the 164 patients was 23–72 years with mean of 44.26 years, and there was no significant difference in age among the cervical cancer, CIN and control groups. All patients or their next of kin provided written informed consent for the collection of samples and subsequent research. The present study was approved by the Ethics Committee of Tongji Medical College, Huazhong University of Science and Technology (IORG0003571).

### Clinical definitions

The medical records of the patients with cervical cancer were reviewed to collect clinical and pathological characters, including age at diagnosis, FIGO stage, pathological type, tumor differentiation, pelvic lymph node metastasis, tumor size, and lymphovascular space invasion (LVSI). The patients were classified according to the revised FIGO staging system for cervical cancer of 2009. The tumor size was the largest diameter of tumor determined by the attending gynecologic oncologist during a pelvic examination preceding surgery. The patients of IA1 stage underwent hysterectomy, and patients between IB1 and IIB underwent radical hysterectomy and pelvic lymph node dissection. Patients with ≥IIB stage underwent radiotherapy or radiotherapy combined with chemotherapy. Stage-specific follow-up after treatment was performed in accordance with the FIGO guidelines. Progression-free survival (PFS) was defined as the time from treatment to the first appearance of tumor recurrence or to the date of last contact. Overall survival (OS) was defined as the time from treatment to death from any cause or to the date of last contact.

### Sample collection, ELISA, and IHC

Serum samples were collected before primary treatment such as surgery, radiation, or chemotherapy. Blood was drawn into serum tubes (10 ml) and centrifuged at 2,200 G/min for 10 min. Serum was harvested, aliquoted and stored at −80 °C until usage.

The concentration of sAng-1 and sAng-2 in serum was assayed by a standardized sandwich enzyme-linked immunosorbent assay (ELISA) in triplicate according to the protocol recommended by the manufacturer (Uscn Life Science Inc., Wuhan, China).

For immunohistochemistry (IHC), paraffin-embedded sections (4 $\mu$m) were hydrated by alcohol and then subjected to antigen retrieval in 10 mM sodium citrate buffer (pH 6.0). After being treated with 3% $H_2O_2$ for 10 min, samples were incubated with primary antibodies overnight at 4 °C. Three primary antibodies were used: human antibodies for detection of Ang-1 (AF923, Polyclonal, Goat IgG), Ang-2 (AF623, Polyclonal, Goat IgG) and CD34 (AF7227, Polyclonal, Sheep IgG) in cervical tissue were all purchased from R&D (Minneapolis, MN, USA); the working concentrations of antibodies were 5 $\mu$g/ml. Standard horseradish peroxidase staining using an appropriate biotinylated secondary antibody (ZSGB-BIO, Peking, PRC), the elite ABC kit (Avidin: Biotinylated enzyme Complex; ZSGB-BIO, Peking, PRC) and diaminobenzidine (ZSGB-BIO, Peking, PRC) were performed according to the manufacturers' instructions. Sections were counterstained with Gill's haematoxylin (Sigma, Aldrich, St. Louis, MO USA). The expression of angiopoietins in epithelial cells and endothelial cells (EC) in cervical tissues were separately evaluated and scored as negative staining and positive staining. MVD was assessed using CD34 (*Yang et al., 2009*).

## Statistical analysis

Statistical analysis was carried out with SPSS 13.0 statistical software (SPSS, Chicago, IL, USA). Continuous variables were expressed as median and range; the data from each category was presented as frequency and percentage. Non-parametric tests were used for comparisons between different groups. The differences in sAng-1, sAng-2, and sAng-1/sAng-2 ratio between groups were analyzed with the Kruskal–Wallis test combined with Dunn's multiple comparison, the Jonckheere–Terpstra test or Mann–Whitney $U$ test, as appropriate. ROC curve was used to analyze the diagnostic value of individual factors. Correlations were analyzed with the Spearman's correlation coefficient. Survival analyses were performed and Kaplan–Meier survival curves were generated; the survival distributions were compared by log-rank test. All tests were two-tailed, and a $P$ value of less than 0.05 was considered to be statistically significant.

## RESULTS

### sAng-2 concentration and sAng-1/sAng-2 ratio are altered in patients with cervical neoplasia

The sAng-1and sAng-2 in 43 patients of normal control, 44 CIN patients and 77 cervical cancer patients (61 CSCC patients and 16 CADC patients) were detected by ELISA. Given that CINs are the precancerous lesions of CSCCs, we first compared the sAng-1, sAng-2 and sAng-1/sAng-2 ratio in the normal, CIN and CSCC patients. We found that the sAng-1 concentration was not significantly different between these three groups, with a median concentration of 58.13 pg/ml (range, 6.79–194.1 pg/ml) in the control group, 28.53 pg/ml (range, 7.22–248.5 pg/ml) in the CIN group and 37.79 pg/ml (range, 1.57–361.2 pg/ml) in the CSCC group (Kruskal–Wallis test, $P = 0.068$, Fig. 1A). However, there was a significant

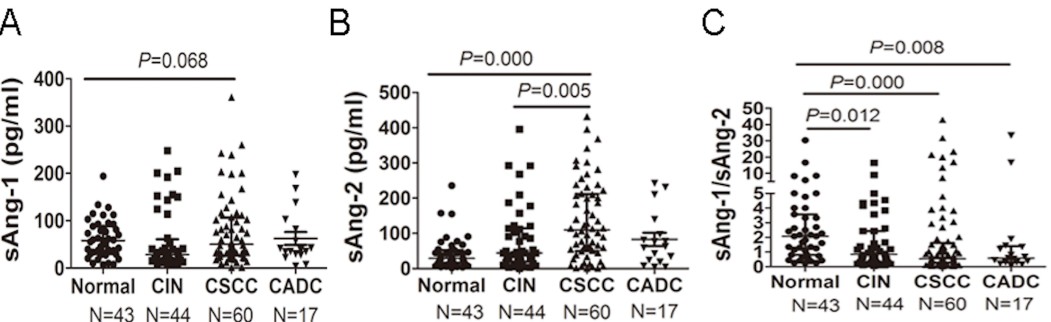

**Figure 1** **sAng-2 concentration and sAng-1/sAng-2 ratio are altered in patients with cervical neoplasia.** sAng-1 and sAng-2 concentrations in 43 patients of normal control, 44 CIN patients, and 77 cervical cancer patients (62 CSCC patients and 15 CADC patients) were determined by ELISA. The differences in Ang-1 (A), sAng-2 (B), and the sAng-1/sAng-2 ratio (C) between groups are shown.

difference in sAng-2 between the three groups (Kruskal–Wallis test, $P = 0.000$), and from normal control (28.87(1.86–235.7) pg/ml), to CIN (44.35(0.70–395.6) pg/ml), then to CSCC (109.90(1.09–431.70) pg/ml), in that with the increasing severity of cervical lesions, the concentration of sAng-2 gradually increased (Jonckheere–Terpstra test, $P < 0.001$; Fig. 1B). The sAng-1/sAng-2 ratio was also significantly different between the three groups (Kruskal–Wallis test, $P < 0.001$; Fig. 1C). The sAng-1/sAng-2 ratio decreased gradually from the control group (median, 2.07; range, 0.25–30.36), to the CIN group (median, 0.85; range, 0.08–17.54), then to the CSCC group (median, 0.54; range, 0.02–42.91) (Jonckheere–Terpstra test, $P < 0.001$; Fig. 1C).

We then compared the concentration of sAng-1 and sAng-2 and the sAng-1/sAng-2 ratio in the CADC group relative to the normal groups. The differences in sAng-1(37.79 vs 58.13 pg/ml) and sAng-2 concentrations (62.97 vs 28.87 pg/m) were not significant (Figs. 1A and 1B). However, the sAng-1/sAng-2 ratio was significantly decreased in CADC patients compared with the normal group (0.58 (0.06–33.33) vs 2.07 (0.25–30.36), $P = 0.008$) (Fig. 1C).

## sAng-2 concentration and the sAng-1/sAng-2 ratio are valuable diagnostic biomarkers for cervical lesions

Having determined the differential sAng-2 concentrations and sAng-1/sAng-2 ratio in cervical cancer lesions compared with normal, we then used ROC curves to test whether the sAng-2 concentration and sAng-1/sAng-2 ratio can serve as biomarkers to distinguish CSCC from normal and CIN. The data of the 43 patients of normal control, the 44 CIN patients, and the 61 CSCC patients were included in the analysis. The area under curve (AUC) of the ROC curves of sAng-2 for discriminating CSCC from normal (Fig. 2A), CSCC from CIN (Fig. 2B), CSCC from normal and CIN (Fig. 2C), CIN and CSCC from normal (Fig. 2D), and invasive cervical cancer from normal (Fig. 2E) were 0.769, 0.662, 0.715, 0.706 and 0.744, respectively. The sAng-2 cutoff value of 48.24 pg/ml provides a high diagnostic accuracy to discriminate CSCC from normal, with a sensitivity of 76.67% and a specificity of 74.42% (Fig. 2F).

The AUC value of sAng-1/ sAng-2 as a diagnostic biomarker for normal to CIN, normal to CSCC, normal to CIN and CSCC and normal to cervical cancer was 0.657, 0.719, 0.693

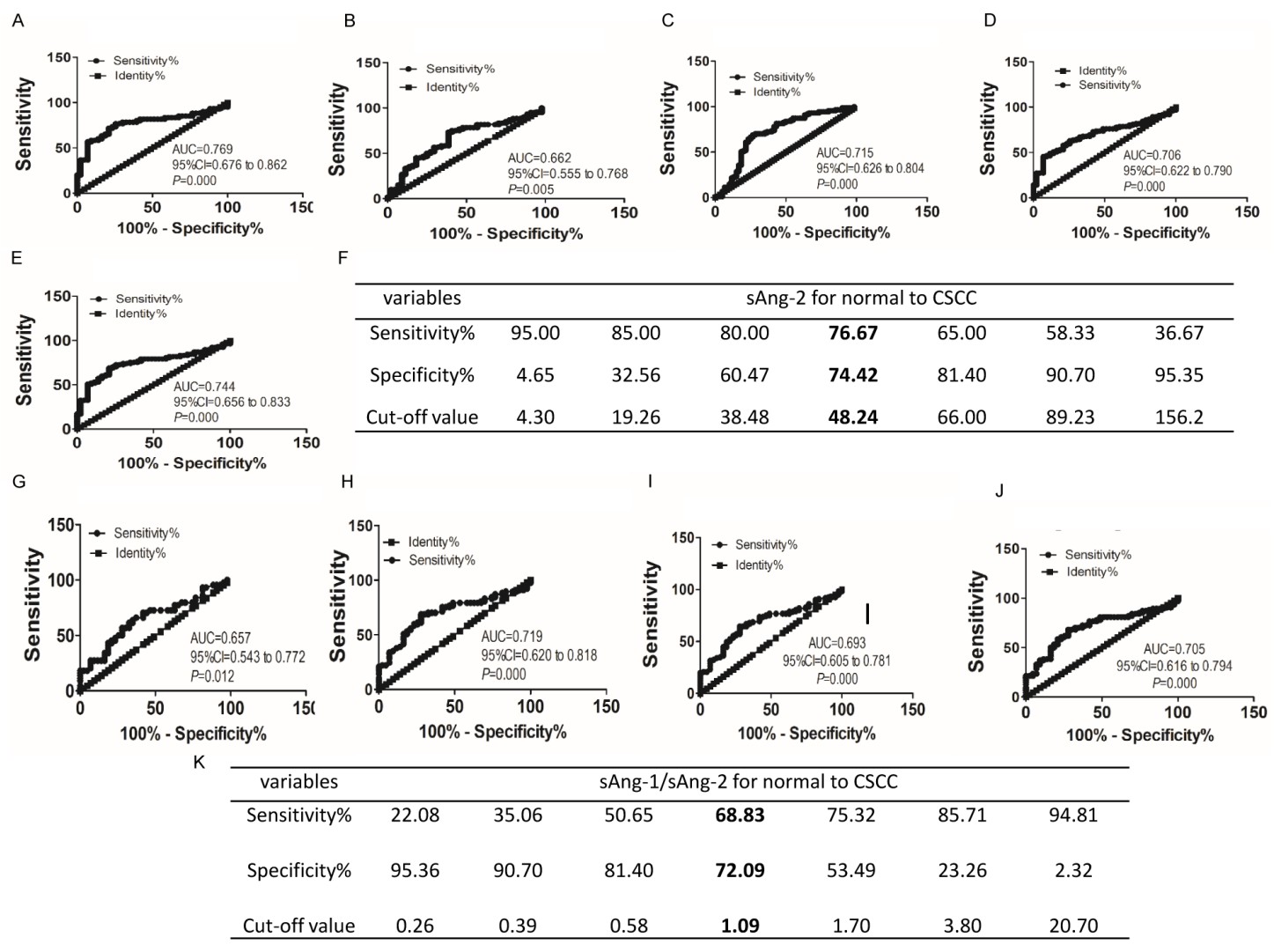

| variables | | sAng-2 for normal to CSCC | | | | | |
|---|---|---|---|---|---|---|---|
| Sensitivity% | 95.00 | 85.00 | 80.00 | **76.67** | 65.00 | 58.33 | 36.67 |
| Specificity% | 4.65 | 32.56 | 60.47 | **74.42** | 81.40 | 90.70 | 95.35 |
| Cut-off value | 4.30 | 19.26 | 38.48 | **48.24** | 66.00 | 89.23 | 156.2 |

| variables | | sAng-1/sAng-2 for normal to CSCC | | | | | |
|---|---|---|---|---|---|---|---|
| Sensitivity% | 22.08 | 35.06 | 50.65 | **68.83** | 75.32 | 85.71 | 94.81 |
| Specificity% | 95.36 | 90.70 | 81.40 | **72.09** | 53.49 | 23.26 | 2.32 |
| Cut-off value | 0.26 | 0.39 | 0.58 | **1.09** | 1.70 | 3.80 | 20.70 |

**Figure 2** **sAng-2 concentration and the sAng-1/sAng-2 ratio are valuable diagnostic biomarkers for cervical lesions.** The ROC curves of sAng-2 for discriminating CSCC from normal (A), CSCC from CIN (B), CSCC from normal and CIN (C), CIN and CSCC from normal (D), and invasive cervical cancer from normal (E). The cutoff values of sAng-2 and corresponding sensitivity and specificity to distinguish CSCC from normal are shown (F). The ROC curves of sAng-1/sAng-2 for discriminating CIN from normal (G), CSCC from normal (H), CIN and CSCC from normal (I) and invasive cervical cancer from normal (J). The cutoff values of sAng-1/ sAng-2 and corresponding sensitivity and specificity to distinguish CSCC from normal are listed (K).

and 0.705 (Figs. 2G–2J), respectively. The cutoff value of 1.09 had a sensitivity of 68.83% and specificity of 72.09% to distinguish CSCC from normal (Fig. 2K).

## sAng-1 and sAng-2 correlate to clinicopathological characteristics in patients with cervical cancer

The correlations between the concentrations of serum angiopoietins and clinicopathological characteristics of cervical cancer were analyzed in a subset of 77 patients with cervical cancer, including 41 stage IA1 to IB1, 20 stage IB2 to IIA and 16 stage ≥IIB based on classification criteria (FIGO, 2009). Characteristics of the patients are summarized in Table 1. We found that sAng-1 concentration was significantly decreased in groups of stage ≥IB2, tumor

**Table 1** The characteristics and serum angiopoietins in patients with cervical cancer ($N = 77$).

| Parameters | Cases (%) |
|---|---|
| Age (year) | |
|     Median (range) | 46 (28–68) |
|     ≤45 | 37 (48.1) |
|     >45 | 40 (51.9) |
| FIGO stage | |
|     IA1–IA2 | 3 (3.9) |
|     IB1 | 38 (49.4) |
|     IB2–IIA2 | 20 (25.9) |
|     IIB | 12 (15.6) |
|     IIIA–IV | 4 (5.20) |
| Pathological type | |
|     Squamous carcinoma | 61 (79.2) |
|     Adenocarcinoma | 16 (20.8) |
| Differentiation | |
|     Well differentiated | 19 (24.7) |
|     Moderately differentiated | 31 (40.2) |
|     Poorly differentiated | 27 (35.1) |
| Pelvic lymph node metastasis | |
|     Negative | 40 (51.9) |
|     Positive | 21 (27.3) |
|     Unknown | 16 (20.8) |
| Tumor size | |
|     ≤2 cm | 52 (67.5) |
|     >2 cm | 25 (32.5) |
| Lymphovascular space invasion | |
|     Negative | 48 (62.3) |
|     Positive | 29 (37.7) |
| sAng-1 (pg/ml) | |
|     Median (range) | 47.81 (1.57–361.21) |
|     High | 14 (18.2) |
|     Low | 63 (81.8) |
| sAng-2 (pg/ml) | |
|     Median (range) | 89.94 (1.09–431.68) |
|     High | 24 (31.2) |
|     Low | 53 (68.8) |
| sAng-1/sAng-2 | |
|     Median (range) | 0.54 (0.02–42.91) |
|     High | 60 (77.9) |
|     Low | 17 (22.1) |

size >2 cm and LVSI compared with stages IA1-IB1, tumor size ≤2 cm, or no LVSI, respectively. sAng-1 concentrations had no significant correlation with differentiation and pelvic lymphatic status. Although the sAng-2 concentration was significantly increased in the poorly differentiated group, no significant correlation with other clinicopathological characteristics were noted. sAng-1/sAng-2 was significantly decreased in stages ≥IB2, poor differentiation and LVSI, but had no significant correlation with pelvic lymphatic metastasis status and tumor size (Table 2).

### sAng-2 concentration positively correlates with Ang-2 expression on epithelia and MVD in cervical tissues

In a subset of 25 patients with invasive cervical cancer (FIGO stage IA2–IIA) and 10 normal control patients, angiopoietin expression and MVD were evaluated in paraffin-embedded cervical tissue samples. Among the 25 cervical cancers, 21 cases were squamous cancer and four were adenocarcinoma; 15 patients had a tumor stage of FIGO IA1 to IB1 and 10 had a tumor stage of FIGO IB2 to IIA; seven tumors were well differentiated (G1), eight were moderately differentiated (G2) and 10 were poorly differentiated cancer (G3); nine patients had lymphovascular space invasion (LVSI); four patients had pelvic lymph node metastasis; the tumor sizes were greater than 2 cm in nine cases.

The IHC assays revealed that both Ang-1 and Ang-2 were highly expressed on tumor cells compared to normal cervical epithelia. The positive expression rate of Ang-1 in cervical cancer epithelia was 60% (15/25) and Ang-2 was 68% (17/25), while normal cervical epithelia had no positive staining. For the expression of Ang-1 and Ang-2 in endothelial cells, there was no significant difference between cervical cancer tissue specimens and normal controls (Figs. 3A and 3B).

MVD, as assessed by CD34 staining, was significantly increased in cervical cancer compared with normal (Fig. 3C). The patients with positive Ang-2 in cervical epithelia had significantly elevated sAng-2 concentration when compared to those with negative Ang-2 expression in cervical epithelia (80.89 (15.54–431.7) pg/ml vs 24.66 (1.86–299.3) pg/ml) ($P = 0.029$, Fig. 3D). In addition, the sAng-1 concentration did not correlate with MVD ($r = 0.256$, $P = 0.138$, Fig. 3E); the sAng-2 concentration positively correlated to MVD ($r = 0.440$, $P = 0.008$, Fig. 3F); the sAng-1/sAng-2 ratio was negatively correlated to MVD ($r = -0.516$, $P = 0.002$, Fig. 3G).

### High sAng-1/sAng-2 ratio predicts poorer prognosis in cervical cancer patients

The prognostic value of sAng-1, -2 and sAng-1/sAng-2 was evaluated in 66 cervical cancer patients with follow-up data. The patients were followed up to October 2016. The median follow-up time was 33.62 months (range, 9–48 months). Of the 66 patients, 17 suffered from recurrence and 8 died. According to the ROC analyses, we choose 131.00 pg/ml, 156.40 pg/ml, and 0.26 as the cut-off values of sAng-1, sAng-2 and sAng-1/sAng-2, respectively, as these values were associated with a specificity of 95% for discriminating between invasive cervical cancer and normal. We found that there were no significant differences in the PFS and OS between the patients with high versus low sAng-1 or sAng-2 (Figs. 4A–4D). However, the sAng-1/sAng-2 ratio was significantly associated with PFS

**Table 2** The correlation between sAng-1, sAng-2, sAng-1/sAng-2 ratio and clinicopathological characteristics in cervical cancer ($N = 77$).

| Variable | Cases | sAng-1 (pg/ml) Median (range) | P | sAng-2 (pg/ml) Median (range) | P | sAng-1/sAng-2 Median (range) | P |
|---|---|---|---|---|---|---|---|
| Age (year) | | | | | | | |
| ≤45 | 37 | 59.92 (1.57–361.21) | 0.266 | 66.28 (1.09–431.68) | 0.203 | 0.77 (0.02–33.33) | 0.101 |
| >45 | 40 | 37.96 (11.84–253.59) | | 113.70 (1.68–368.08) | | 0.46 (0.08–50.00) | |
| FIGO stage | | | | | | | |
| IA1-IB1 | 41 | 59.92 (1.57–361.21) | **0.046** | 71.38 (1.19–431.68) | 0.691 | 0.85 (0.02–23.35) | **0.010** |
| ≥IB2 | 36 | 36.80 (1.91–253.59) | | 94.01 (1.09–306.77) | | 0.37 (0.04–42.91) | |
| Pathological type | | | | | | | |
| Squamous carcinoma | 61 | 48.86 (1.57–361.21) | 0.920 | 101.99 (1.09–431.68) | 0.085 | 0.53 (0.02–50.00) | 0.444 |
| Adenocarcinoma | 16 | 39.63 (3.68–197.66) | | 51.92 (4.37–242.26) | | 0.69 (0.06–33.33) | |
| Differentiation | | | | | | | |
| Well and moderately differentiated | 50 | 51.90 (3.68–361.21) | 0.364 | 64.31(1.09–431.68) | **0.002** | 0.88 (0.06–31.71) | **0.000** |
| Poorly differentiated | 27 | 42.47(1.57–260.11) | | 152.49 (5.91–368.08) | | 0.27 (0.02–42.91) | |
| Pelvic lymph node metastasis | | | | | | | |
| Negative | 40 | 81.68 (9.52–253.59) | 0.421 | 65.70 (1.09–431.68) | 0.476 | 0.78 (0.14–42.91) | 0.281 |
| Positive | 21 | 45.14 (3.68–361.21) | | 83.67 (1.68–395.47) | | 0.77 (0.02–23.35) | |
| Tumor size | | | | | | | |
| ≤2 cm | 52 | 59.92 (1.57–361.21) | **0.034** | 101.99 (1.19–431.68) | 0.306 | 0.58 (0.02–42.91) | 0.286 |
| >2 cm | 25 | 34.41 (1.91–203.59) | | 69.23 (1.09–291.58) | | 0.35 (0.04–31.71) | |
| Lymphovascular space invasion | | | | | | | |
| Negative | 48 | 56.12 (1.57–361.21) | **0.039** | 80.66 (1.19–431.68) | 0.328 | 0.71 (0.02–50.00) | **0.013** |
| Positive | 29 | 38.12 (1.91–121.64) | | 114.26 (1.09–299.27) | | 0.41 (0.04–20.00) | |

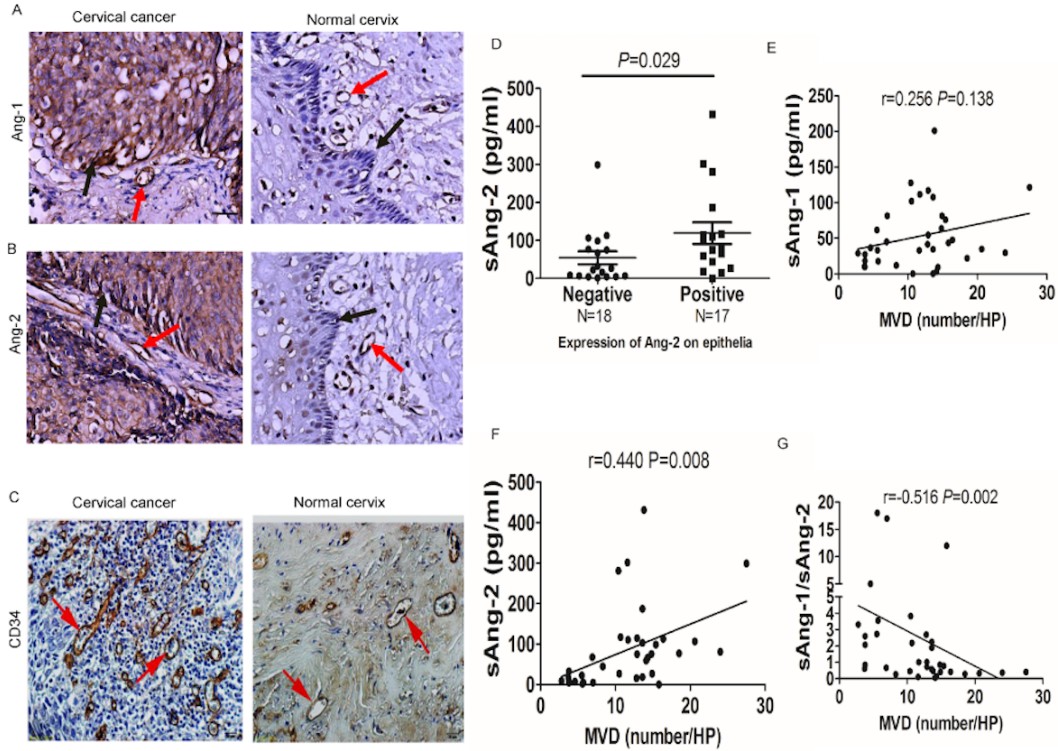

**Figure 3   sAng-2 concentration positively relates with Ang-2 expression on the epithelia and MVD in cervical tissues.** Representative immunohistochemical staining of Ang-1 (A), Ang-2 (B) and CD34 (C) in 25 cervical cancer tissue specimens and 10 normal controls. Black arrows denote positively stained epithelial cells, whereas red arrows denote positively staining endothelial cells, all appearing brown. Scale bar, 20 μm. (D) sAng-2 is significantly higher in the patients with positive Ang-2 expression on cervix epithelia than those with negative Ang-2 expression. The scatter diagrams show the correlations of sAng-1 (E), sAng-2 (F) and sAng-1/ sAng-2 ratio (G) to MVD in the 35 cervical tissue specimens.

(Fig. 4E, $P = 0.046$) and OS (Fig. 4F, $P = 0.040$) in cervical cancer patients. In the low sAng-1/sAng-2 ratio group, 7 of 14 (50.0%) patients developed recurrent disease and 4 (28.6%) patients died. The estimated 3-year PFS and OS were 42.86% and 71.43%, respectively. In the high sAng-1/sAng-2 ratio group, 10 of 52 (19.23%) patients had recurrence and 4 (7.7%) patients died, resulting in an estimated 3-year PFS and OS of 78.96% and 91.55%, respectively.

## DISCUSSION

In the present study, we found that the concentration of sAng-2 was significantly increased in patients with cervical cancer compared with normal controls and CIN patients and the sAng-1/sAng-2 ratio was decreasing during the malignant transformation of cervical epithelia. Moreover, the sAng-2 and the sAng-1/sAng-2 ratio exhibit encouraging accuracies for diagnosis of a CSCC. In patients with cervical cancer, low sAng-1/sAng-2 ratio was significantly associated with advanced stage, poor differentiation, LVSI, high MVD and poor survival. These findings suggest that the ratio of serum Angiopoietin-2 to Angiopoietin-1 is a valuable diagnostic and prognostic biomarker in patients with cervical cancer.

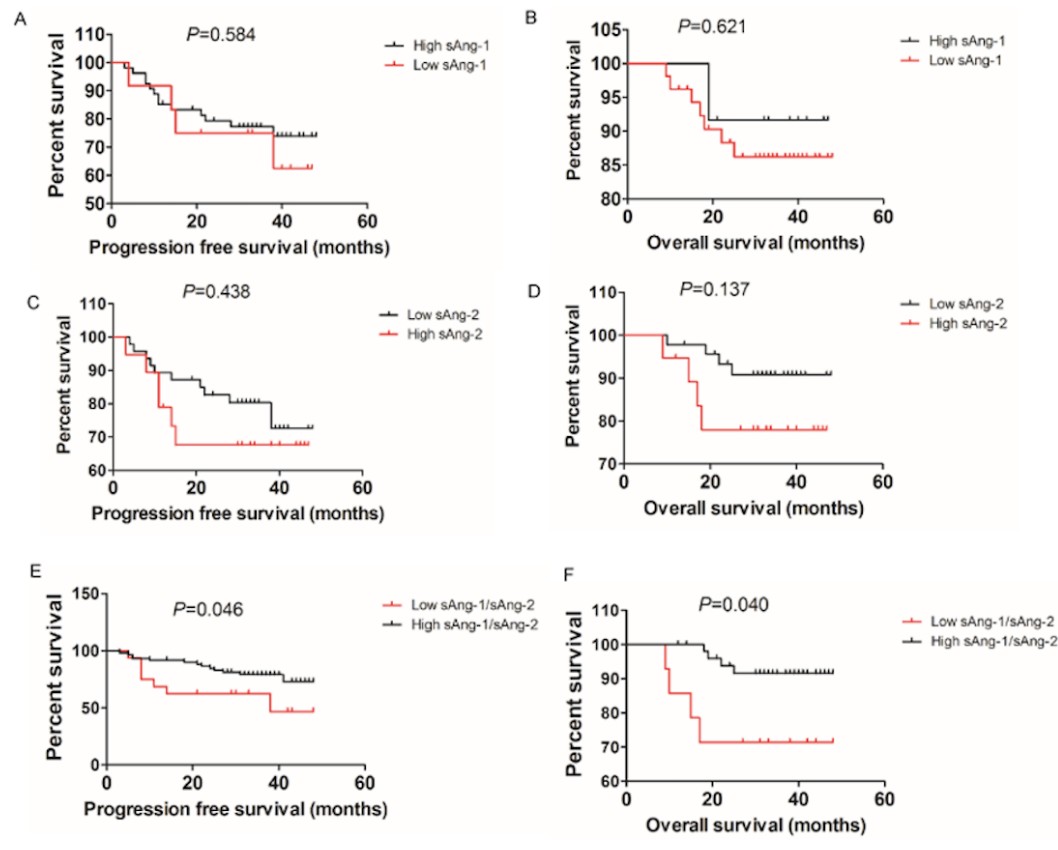

**Figure 4** **High sAng-1/sAng-2 ratio predicts poorer survival in cervical cancer patients.** Kaplan–Meier survival analysis of the progression-free survival and the overall survival among 66 cervical cancer patients stratified by sAng-1 (A, B), sAng-2 (C, D) and sAng-1/sAng-2 ratio (E, F).

Tumor angiogenesis is the most fundamental trait for tumor growth. Tumor nodules cannot derive nutrients through diffusion when greater than 1–2 mm$^3$ (*Hanahan & Weinberg, 2011*; *Jain & Carmeliet, 2012*). Previous reports on various type of cancer, such as colorectal cancer, gastric carcinoma, and glioma, have shown that elevated Ang-2 in these tumor cells resulted in increased MVD and stimulated tumor angiogenesis (*Augustin et al., 2009*; *Fagiani & Christofori, 2013*; *Shim, Ho & Wong, 2007*). Moreover, in melanoma, tumor cells secreted sAng-2 and expressed Tie-2 suggesting that Ang-2 may act as an autocrine regulator of tumor cell migration and invasion (*Helfrich et al., 2009*). We found that Ang-2 was expressed in tumor cells of cervical cancer tissues. The sAng-2 concentration also positively correlated with Ang-2 expression in epithelia and MVD of cervical tissues, suggesting that the concentration of sAng-2 may reflect the activity of blood vessel formation and soluble Ang-2 secreted by cervical cancer cells may be the main source of serum sAng-2 of cervical cancer. Given that the sAng-2 and the sAng-1/sAng-2 ratio were significantly associated with MVD in cervical cancer (Figs. 3F–3G), they might be used to evaluate the response of tumor to anti-angiogenesis therapy in real-time. In metastatic colorectal cancer, sAng-2 was a candidate biomarker for outcome of patients treated with bevacizumab-containing therapy (*Goede et al., 2010*). Very recently, a study

indicated bevacizumab reduced mortality by 30% in patients with recurrent, persistent, or metastatic cervical cancer when combined with chemotherapy compared with those who receive chemotherapy alone (*Penson et al., 2015*). The utility of sAng-2 and sAng-1/sAng-2 ratio as biomarkers to predict outcome of bevacizumab-containing treatment in advanced cervical cancer should be further validated.

We found that sAng-2 concentration gradually increased and the ratio of sAng-1/ sAng-2 gradually decreased with the severity of cervical lesion from control to CIN to CSCC. The sAng-1/sAng-2 ratio was also decreased in CADC patients compared with normal group. In a previous study, Kopczynska et al. assessed plasma concentrations of Ang-1and Ang-2 in 34 patients with cervical cancer and 20 healthy volunteers. They reported that plasma concentrations of Ang-1 and Ang-2 were significantly higher in cervical cancer patients than in controls. Moreover, the plasma Ang-1/Ang-2 ratios was increased in cervical cancer compared with control and was higher in stage I than in stage II–III (*Kopczynska et al., 2009*). The increase in circulating Ang-2 in patients with cervical cancer was confirmed by our findings. However, the change in Ang-1/Ang-2 ratio reported by Kopczynska et al. was opposite to that shown in our study and we did not find significant differences in Ang-1 concentration between control, CIN, and cervical cancer. These inconsistencies may be related to the differences in study population and the materials used in ELISA. In addition, the influence of anticoagulants used during blood collection on plasma angiopoietin detection cannot be excluded.

The ability of sAng-2 concentration and the ratio of sAng-1 to sAng-2 to distinguish between normal, CIN, and cervical cancer was similar to results reported in epithelial ovarian cancer, where the area under the curve for serum Ang-2 in ROC analysis was 0.75 to differentiate ovarian cancer from benign or borderline ovarian tumors (*Sallinen et al., 2014*; *Sallinen et al., 2010*). In melanoma, use of 90% quantile of healthy controls as a cutoff value could discriminate well between control individuals and individuals with melanoma stage III or IV patients (*Helfrich et al., 2009*). sAng-2 was elevated in patients with hepatocellular carcinoma, suggesting the potential use of angiopoietin-2 as a valuable diagnostic biomarker for the detection of hepatocellular carcinoma (*Bouattour, Payance & Wassermann, 2015*). Our findings suggest that sAng-2 concentration and the ratio of sAng-1 to sAng-2 may be novel diagnostic biomarkers for the detection of normal, CIN and cervical cancer. Additional research to determine cutoff values based on population characteristics are being explored.

Both sAng-1 and sAng-2 have been found to be involved in tumor growth and progression. In most conditions, elevated sAng-1 played anti-tumorigenic effects, whereas upregulated sAng-2 stimulated tumor growth and progression (*Augustin et al., 2009*; *Eklund & Saharinen, 2013*; *Fagiani & Christofori, 2013*; *Shim, Ho & Wong, 2007*). In some tumors, such as glioblastoma, breast cancer, hepatocellular carcinoma and pancreatic cancer, both sAng-1 and sAng-2 are upregulated ((*Augustin et al., 2009*; *Fagiani & Christofori, 2013*; *Shim, Ho & Wong, 2007*; *Eklund & Saharinen, 2013*). Elevated sAng-1 was found in lung cancer and epithelial ovarian cancer, yet no relation was found with diagnosis and prognosis of tumor (*Park et al., 2009*; *Sallinen et al., 2014*). On the contrary, patient with papillary thyroid cancer were found to have significantly decreased sAng-1 compared with the

normal control (*Makki et al., 2013*). We found that sAng-2, but not sAng-1 was increased, suggesting that factors driving tumor angiogenesis may be unique to tumor types.

Elevated sAng-2 tightly correlated with lymphatic and liver metastasis and shorter survival in pancreatic ductal adenocarcinoma (*Schulz et al., 2011*) and neuroendocrine tumor patients (*Detjen et al., 2010*; *Helfrich et al., 2009*). sAng-2 has also been suggested to be a useful biomarker in melanoma as progression and metastasis correlated with tumor load and overall survival (*Helfrich et al., 2009*). In epithelial ovarian cancer, the serum concentration of Ang-2 predicted poor overall survival (*Sallinen et al., 2014*; *Sallinen et al., 2010*). In our study, we found that lower sAng-1 in patients with cervical cancer correlated with more advanced cervical cancer, including higher stage, larger tumor size and LVSI. We also found higher sAng-2 levels in patients with poorly differentiated cervical cancer. Overall, the ratio of sAng-1/sAng-2 was predictive of PFS and OS, showing that patient with cervical cancer and a lower sAng-1/sAng-2 had a poorer prognosis.

The present study assessed for the first time the serum concentrations of angiopoietins and sAng-1/sAng-2 ratio in healthy women and patients with CIN and invasive cervical cancer and evaluated the diagnostic and prognostic power of sAng-1, sAng-2 and sAng-1/sAng-2 in cervical cancer. A major limitation of this study was the small sample size of cervical cancer patients ($N = 77$), particularly that of CADC ($N = 16$), which may be associated with selection bias and low statistical power. In addition, the follow-up periods of the patients were relatively short. Multicenter studies with long-term follow-up are needed to verify our findings.

## CONCLUSION

In summary, sAng-2 was significantly increased in cervical cancer patients and a low sAng-1/sAng-2 ratio was highly effective of discriminating cervical cancer patients from normal control and predictive of poor survival in patients with cervical cancer. These findings suggest that the sAng-1/sAng-2 ratio is a potential diagnostic and prognostic biomarker for cervical cancer. Further studies to validate these findings and explore the utility of anti-angiogenesis therapeutic targets for treatment of advanced or recurrent cervical cancer are needed.

**Abbreviation**

| | |
|---|---|
| **Ang** | angiopoietin |
| **sAng-2** | serum Ang-2 |
| **VEGF** | vascular endothelial growth factor |
| **CIN** | cervical intraepithelial neoplasia |
| **MVD** | microvessel density |
| **CSCC** | cervical squamous cell carcinoma |
| **CADC** | cervical adenocarcinoma |
| **LVSI** | lymphovascular space invasion |
| **PFS** | Progression-free survival |
| **OS** | Overall survival |
| **ELISA** | enzyme-linked immunosorbent assay |

| HRP | Horseradish Peroxidase |
|-----|------------------------|
| IHC | immunohistochemistry |
| DAB | diaminobenzidine |
| EC | endothelial cells |
| HP | high power field |

### Funding

This work was supported by the National Natural Science Foundation of China (No. 81472443 and 81572572). The funders had no role in study design, data collection and analysis, decision to publish, or preparation of the manuscript.

### Grant Disclosures

The following grant information was disclosed by the authors:
National Natural Science Foundation of China: 81472443, 81572572.

### Competing Interests

The authors declare there are no competing interests.

### Author Contributions

- Ping Yang conceived and designed the experiments, performed the experiments, analyzed the data, wrote the paper, prepared figures and/or tables, reviewed drafts of the paper.
- Na Chen and Dongyun Yang performed the experiments.
- Janet Crane wrote the paper, reviewed drafts of the paper.
- Shouhua Yang analyzed the data, wrote the paper.
- Hangyu Wang, Ruiqing Dong, Xiaoqing Yi, Lisha Xie and Guo Jing contributed reagents/materials/analysis tools.
- Jing Cai conceived and designed the experiments, analyzed the data, wrote the paper, prepared figures and/or tables, reviewed drafts of the paper.
- Zehua Wang conceived and designed the experiments, reviewed drafts of the paper.

### Human Ethics

The following information was supplied relating to ethical approvals (i.e., approving body and any reference numbers):

The Ethics Committee of Tongji Medical College, Huazhong University of Science and Technology approval to carry out the study within its facilities (IORG No:IORG0003571). All patients or their next of kin provided written informed consent for the collection of samples and subsequent research.

### Data Availability

The raw data has been supplied as Data S1.

## Supplemental Information

Supplemental information for this article can be found online at http://dx.doi.org/10.7717/peerj.3387#supplemental-information.

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
