# Peer review of "The ratio of serum Angiopoietin-1 to Angiopoietin-2 in patients with cervical cancer is a valuable diagnostic and prognostic biomarker"

_PeerJ, doi:10.7717/peerj.3387_

## Round 0.1 · original submission · Major Revisions

The manuscript was reviewed by 3 experts in the field and the summary of thier comments suggest that it can be cosidered for publication following an extensive revision that will adress all the comments raised by the reviewers

Reviewer 1 ·

Basic reporting

No comment on all four sections

Experimental design

Two thumbs up for all four sections, though readers practicing basic science research might require a more detailed description of laboratory methods

Validity of the findings

Authors managed to find statistically significant results in a relatively small group of patients. However, the were not dazzled by their success and emphasized the need for further and larger studies to validate the specificity of their findings for cervical cancer

Additional comments

Authors presented results of the study in the materials and methods section describing in detail patients characteristics. Particularly, lines 123-128 on page 10 and lines 137-145 describing the patient population should be transferred to the results section

Reviewer 2 ·

Basic reporting

Review
The ratio of serum Angiopoietin-1 to Angiopoietin-2 in patients with cervical cancer is a valuable diagnostic and prognostic biomarker
This prospective study was conduct in order to investigate the diagnostic and prognostic values of serum angiopoietin 1 and 2 (sAng-1 and sAng-2) and their ratio in cervical cancer,as well as to assess the microvessel density and investigate their correlation with the circulating angiopoietin. Serum samples were obtained from 164 patients- The sAng-1 and sAng-2 concentrations were analyzed in 77 women with cervical cancer, 44 women with cervical intraepithelial neoplasia (CIN) and 43 women without cervical lesions.
This study demonstrated that decreased sAng-1/sAng-2 ratio was significantly associated with advanced tumor stage, poor differentiation, LVSI and high-micro-vessel density in cervical cancer. The authors suggested that sAng-1/sAng-2 ratio is a valuable diagnostic and prognostic biomarker in patients with cervical cancer.

Experimental design

There are few comments regarding this study-
• The article contain all of the components.
• Abstract- is not well-organized .The abstract should be organized according to this next order-objective, Methods, results and conclusions.
• Introduction-in the last paragraph the author should write about the aim of their study. Any other information regarding the study population or regarding the conclusions should not be written in this section.
• Materials and methods-
1) This section is not well organized as well.it should be organized according to this next order - Study groups and inclusion criteria, Clinical definitions, Sample collection and ELISA/IHC and Statistical analysis.
2) In the first ans second paragraph the demographic/clinical information about the patients should be written in the results section (Table 1).
3) The information about the type of the research and the information about the informed consent should be written in the study groups and inclusion criteria section.
4) The section about the ELISA and the IHC is too detailed and should be shorter.

Validity of the findings

• Materials and methods-
1) This section is not well organized as well.it should be organized according to this next order - Study groups and inclusion criteria, Clinical definitions, Sample collection and ELISA/IHC and Statistical analysis.
2) In the first ans second paragraph the demographic/clinical information about the patients should be written in the results section (Table 1).
3) The information about the type of the research and the information about the informed consent should be written in the study groups and inclusion criteria section.
4) The section about the ELISA and the IHC is too detailed and should be shorter.
• Results –
1) The first table should be on demographic criteria
2) In the beginning of each paragraph should be information about the number of patient in each group.
3) What about The comparison between the concentration of sAng-1 and sAng-2 and the ratio between them in cases of CADC and AGC ?
4) The different between CSCC and invasive cancer (paragraph 3.2) is not well understood.

Additional comments

There are few comments regarding this study-
• The article contain all of the components.
• Abstract- is not well-organized .The abstract should be organized according to this next order-objective, Methods, results and conclusions.
• Introduction-in the last paragraph the author should write about the aim of their study. Any other information regarding the study population or regarding the conclusions should not be written in this section.
• Materials and methods-
1) This section is not well organized as well.it should be organized according to this next order - Study groups and inclusion criteria, Clinical definitions, Sample collection and ELISA/IHC and Statistical analysis.
2) In the first ans second paragraph the demographic/clinical information about the patients should be written in the results section (Table 1).
3) The information about the type of the research and the information about the informed consent should be written in the study groups and inclusion criteria section.
4) The section about the ELISA and the IHC is too detailed and should be shorter.
• Results –
1) The first table should be on demographic criteria
2) In the beginning of each paragraph should be information about the number of patient in each group.
3) What about The comparison between the concentration of sAng-1 and sAng-2 and the ratio between them in cases of CADC and AGC ?
4) The different between CSCC and invasive cancer (paragraph 3.2) is not well understood.
• Discussion – this section is also not well organized-
1. The author should start with few sentences regarding the main findings in his study.
2. The last paragraph should be in the beginning of the discussion.
3. There are few places in which the references are missing.
4. A paragraph regarding the Strengths and limitations of the study is lacking.
5. The conclusion section should be shorter.

Reviewer 3 ·

Basic reporting

This manuscript investigates the diagnostic and prognostic values of serum angiopoietin 1 and 2 and their ratio in cervical cancer.
The manuscript can be improved by addressing the following comments:

1. Abstract – please spell out ROC
2. Introduction – line 98 – I would like to ask the authors to mention that Ang-1, Ang-2, Ang-1/Ang-2 ration, and Tie-2 were significantly higher in cervical cancer patients (not just “altered”) in a previous similar study (Kopczynska et al., 2009).
3. Line 110 – The last sentence of the introduction can be moved at the beginning of the Discussion.
4. Materials and methods – please be consistent in using the same stage (e.g. “Ib1” or “IB1”).
5. Line 166 – please spell out O.D.

Experimental design

6. Results – Please correct all “P=0.000” throughout the paper (text, figures, and table); P-value can be shown as <0.001, but not zero.
7. Line 210 – can you please recheck the range for the CSCC group? In the text the range is 3.68-197.66 pg/ml, but the figure shows the highest value as more than 300 pg/ml.
8. Please add the units after medians and range (lines 218, 219, 223, 225)
9. Please recheck the range for CSCC group (line218-219) and CADC group (line 225); they are the same in the text.
10. Line 240 – please use “and” instead of “&”.
11. Line 256 – please change “relates” with “correlates”.
12. Table 1 – please change “rage” with “range” (in all three columns) and “&” with “and” (in the “differentiation” section).
13. Table 1 – FIGO stage – I would like to ask the authors if they have seen any significant differences between serum angiopoietins and different stages (I-II vs. III-IV or I vs. II vs. III vs. IV). Is there a reason the authors chose only IA1-IB1 vs. ≥IB2? Also, is there any difference between the FIGO stages and the pathological types?
14. Please add “HP” (from figure 3 E, F, G) in the abbreviation list.
15. Can the authors please explain why all the results presented in figure 3 are important to this paper? The importance of these results is not well explained in the discussion and they seem to be more relevant to some data not shown in this manuscript (lines 357-363).

Validity of the findings

15. Discussion – I would like to ask the authors to start the discussion by summarizing the findings of this study.
16. Line 336 – please rephrase – the authors did not demonstrate that sAng-2 was highly expressed in patients with poorly differentiated cervical cancer; they reported higher serum Ang-2 levels in patients with poorly differentiated cervical cancer.
17. Line 341 – please remove “with” before “treated”.
18. Line 357, 360 – Please change “data not show” with “data not shown”.
19. Line 341-344 – please provide the reference.
20. I believe the discussion section needs to be improved. In the discussion there is no mention about the previous study by Kopczynska et al., 2009, which studied plasma levels of angiopoietins and their levels; how are the findings of this manuscript related to the previous study? Is there a difference between the levels of angiopoietins found in the previous study and this study (plasma vs. serum levels)?

---

## Round 0.2 · accepted · Accept

Both the reviewers and I have found your manuscript suitable for publication. Therefore, your manuscript is accepted for publication in the Journal

Reviewer 1 ·

Basic reporting

Corrected appropriately

Experimental design

Corrected appropriately

Validity of the findings

Corrected appropriately

Additional comments

Authors have addressed and corrected the problematic points as presented by two other reviewers and myself
I would recommend to accept the article

Reviewer 3 ·

Basic reporting

The authors addressed all the comments of the reviewers and the manuscript was improved.

Experimental design

No comment.

Validity of the findings

No comment.